# Deposition temperature-mediated growth of helically shaped polymers and chevron-type graphene nanoribbons from a fluorinated precursor

Jacob D. Teeter[1,5], Mamun Sarker [2,5], Wenchang Lu[3], Chenggang Tao[1,4], Arthur P. Baddorf [1], Jingsong Huang [1] ✉, Kunlun Hong [1], Jerry Bernholc[3], Alexander Sinitskii [2] & An-Ping Li [1,4] ✉

Graphene nanoribbons (GNRs) of precise size and shape, critical for controlling electronic properties and future device applications, can be realized via precision synthesis on surfaces using rationally designed molecular precursors. Fluorine-bearing precursors have the potential to form GNRs on nonmetallic substrates suitable for device fabrication. Here, we investigate the deposition temperature-mediated growth of a new fluorine-bearing precursor, 6,11-diiodo-1,4-bis(2-fluorophenyl)-2,3-diphenyltriphenylene ($C_{42}H_{24}F_2I_2$), into helically shaped polymer intermediates and chevron-type GNRs on Au(111) by combining scanning tunneling microscopy, X-ray photoelectron spectroscopy, and density functional theory simulations. The fluorinated precursors do not adsorb on the Au(111) surface at lower temperatures, necessitating an optimum substrate temperature to achieve maximum polymer and GNR lengths. We compare the adsorption behavior with that of pristine chevron precursors and discuss the effects of C-H and C-F bonds. The results elucidate the growth mechanism of GNRs with fluorine-bearing precursors and establish a foundation for future synthesis of GNRs on nonmetallic substrates.

Graphene nanoribbons (GNRs) feature moderate and tunable band gaps due to quantum confinement and edge effects[1], distinguishing them from graphene and rendering them well suited for a range of nanoscale device applications[2–5]. Precision syntheses of GNRs with chemically defined width, periodicity, topology, doping, and functionalization are crucial for realizing their intended properties[6–10]. Within the realm of topological variations, the studies of chevron-type GNRs have garnered significant attention ever since their initial report by Cai et al.[11] A gamut of chevron-type GNRs has been synthesized in a bottom-up manner on several metal surfaces under ultra-high-vacuum conditions[12–17], as well as via large-scale chemical vapor deposition[18] and solution-phase processes[19–21]. As band-gap engineering of GNRs has evolved from width and geometry modulation[1,22,23] and heteroatomic substitution and functionalization[24–26] to topological phases[27–30] and spin control[31], it is becoming ever more desirable to be able to fine-tune the electronic characteristics of the GNRs in question.

Unlike nitrogen and sulfur dopings[15,16,18,32], which are used primarily to tune the electronic properties of GNRs, fluorine (F)-substitution is utilized not only to adjust the electronic properties[14,33] but also to facilitate inter- and intra-molecular aryl–aryl coupling through HF elimination on both metal and metal oxide surfaces[34,35]. Careful precursor design allows F atoms to be located out of the region of cyclodehydrogenation such that F is retained in the fully cyclized GNRs[14]. However, fluorine atoms close to the cyclodehydrogenation region can undergo C-F bond scission during the reaction process, resulting in non-functionalized GNRs[33]. C-F bond cleavage can be particularly harnessed to grow carbon-based nanostructures directly on metal oxides[36,37]. For instance, it was recently demonstrated that F-bearing precursors can be used in a new approach based on HF elimination to grow atomically precise GNRs on rutile $TiO_2$, a nonmetallic substrate suitable for device fabrications[38,39]. The F-substitution in a chevron precursor thus provides a promising avenue to synthesize GNRs with enhanced diversity and tunability of electronic properties.

A first step in this regard is optimizing the self-assembly and growth of the polymers and GNRs derived from the F-bearing chevron precursors. Owing to the many functional groups or heteroatoms with which precursors

[1]Center for Nanophase Materials Sciences, Oak Ridge National Laboratory, Oak Ridge, TN, USA. [2]Department of Chemistry, University of Nebraska-Lincoln, Lincoln, Nebraska, USA. [3]Department of Physics, North Carolina State University, Raleigh, NC, USA. [4]Department of Physics and Astronomy, University of Tennessee, Knoxville, TN, USA. [5]These authors contributed equally: Jacob D. Teeter, Mamun Sarker. ✉e-mail: huangj3@ornl.gov; apli@ornl.gov

can be equipped, appropriate conditions are required for enhancing the coverage and length of GNRs on a substrate. We present here growth conditions that facilitate and inhibit the growth of chevron-type GNRs starting from a new precursor bearing internal fluorine atoms. Additionally, we compare the adsorption behavior to that of pristine chevron precursors, analyzing the effects of C-H and C-F bonds. Our results contribute to a deeper understanding of GNR synthesis and pave the way for future advancements in fabricating GNRs on nonmetallic substrates.

## Results and discussion

We synthesized a new molecular precursor, 6,11-diiodo-1,4-bis(2-fluorophenyl)-2,3-diphenyltriphenylene ($C_{42}H_{24}F_2I_2$, **1**), with fluorine substitution placed at internal ortho-positions. The synthetic details are provided in the Supplementary Methods, with additional nuclear magnetic resonance (NMR) spectra presented in Supplementary Figs. 1-7. This precursor has the potential to form chevron-type GNRs on a nonmetallic substrate suited for device fabrication but here we explore the effects of fluorination on GNR growth on Au(111) surface, as illustrated in Fig. 1, by following a two-step thermally triggered on-surface reaction process.

We find that adsorption of the precursor **1** to the substrate is largely dependent upon the temperature of the substrate during deposition. After deposition with the substrate at room temperature, molecules are only observed along the step edges (Fig. 2a, b). Organic molecules like precursor **1** have been reported to selectively decorate step edges[40,41]. The precursors **1** that do adsorb form very short oligomers, with individual and bright lobe-like features decorating the step edge which we attribute to the lost iodine atoms (Fig. 2b, green circle). Keeping the substrate at 100 °C during deposition is sufficient to consistently deiodinate the precursors and allows assembly on the Au(111) terraces as observed in Fig. 2c, d. However, the effectiveness of the deiodination, polymerization, and assembly processes seems to be enhanced at higher temperatures such as 180 °C (Fig. 2e, f), where much greater coverage and average polymer length are achieved. This behavior is in contrast to previously reported behavior for pristine brominated or iodinated chevron precursors on Au(111)[42] (see Supplementary Fig. 8 as well).

Polymers of **1** are observed to be similar in appearance to previously reported STM images of non-functionalized chevron-type polymers[11,12] (see Supplementary Fig. 9)—both poly-**1** and previously reported chevron polymers appear to adopt a π-π interaction-stabilized assembly mode which favors the growth of small 2D islands[43]. However, a new helical feature is identified for the polymer intermediates from both experiments and density functional theory (DFT) calculations (Supplementary Figs. 10 and 11). To also discern the relative orientations of the fluorine atoms in the polymers, i.e., towards or away from the surface, we have calculated the energies and simulated STM images of these possible configurations, further discussed below and shown in Supplementary Fig. 12, respectively. The GNRs arising from the cyclization step demonstrate no structural features to be attributed to lingering fluorine atoms, unlike edge-fluorinated GNRs[14], suggesting that the expected HF

elimination has occurred by the time cyclodehydrogenation and cyclodehydrofluorination are complete.

In order to better understand the behavior of the precursor **1** upon deposition—particularly if polymers and GNRs retained the monomers' fluorine functionalization—we employed X-ray photoelectron spectroscopy (XPS) to characterize a polymer sample grown at 180 °C on an Au(111)/mica thin film. Figure 3 shows the results of these experiments. This sample was transferred from the growth chamber of the STM system, in air, to a separate chamber for the XPS measurements. Measurements were acquired immediately upon introduction to the XPS chamber, 72 h afterwards, and then upon annealing to 400 °C. This was performed to acquire information regarding the short-term air and vacuum stability of the polymer sample as well as whether fluorine atoms survived through the polymerization and GNR formation steps.

As can be seen in Fig. 3, the sample presents clear signatures of fluorine in the form of the F 1s peak and the shoulder adjacent to the C 1s peak, attributed to the carbon in $C_2F$, utilized by Panighel et al. as a model for a fluorinated benzene moiety[14,44]. The I $3d^{5/2}$ peak is also observed in the polymer sample at room temperature, consistent with our STM results as well as previously reported XPS results[45], where iodine was found to persist in the form of Au-I up to approximately 350 °C. Annealing to 400 °C induces the expected cyclodehydrogenation and cyclodehydrofluorination reactions in the fluorinated polymers, resulting in chevron-type GNRs and desorption of iodine-attributed features from the surface. In the XPS spectra, this is marked by the disappearance of both the I $3d^{5/2}$ and F 1s peaks. The $C_2F$ shoulder in the C 1s peak is likewise lost in this transition, which supports our conclusion that the chevron GNRs observed in STM possessed no lingering fluorine atoms. Additional individual elemental spectra can be found in Supplementary Figs. 13-16.

Here we note that the polymer intermediates are observed to manifest a helical conformation. This new feature of helicity, a special case of axial chirality, has not been reported in previous bottom-up synthesis of chevron-type GNRs[11,15,16,18]. Consistent with earlier STM images of non-functionalized chevron-type polymers[11,12,43], each polymer chain is visualized as a "stripe" shape, featuring low-lying triphenylene cores flanked by periodic bright regions attributed to the peripheral phenyl rings. Upon closer examination, these polymers in fact exhibit distinct helical signatures on the phenyl side groups. As shown in Fig. 4, the line profiles along the top and bottom fringes of a representative polymer chain, where the peripheral phenyl rings are situated, display asymmetric low-high and high-low features, respectively, unequivocally indicating a helical conformation. The precursor **1** can polymerize into chains with four possible helicities, including P- and M-, meso- and nonhelical conformations (Supplementary Fig. 11). By comparing the experimental line profiles with those generated from our DFT calculations, we find that the observed polymer intermediates belong to a P-conformation (Supplementary Fig. 11a, b) instead of an M-conformation (Supplementary Fig. 11c, d). In contrast, meso-chains would show asymmetric high-low feature for both traces (Supplementary Fig. 11e, f), while nonhelical chains would show two peaks of equal heights in

**Fig. 1 | Synthetic growth process for polymers and GNRs from the fluorine-bearing precursor 1, 6,11-diiodo-1,4-bis(2-fluorophenyl)-2,3-diphenyltriphenylene ($C_{42}H_{24}F_2I_2$), on Au(111).** Fluorine-bearing iodinated precursor **1** forms helically shaped polymers (poly-**1**) on Au(111) with increasing coverages from room temperature (RT) to 100 °C and 180 °C. Heating the polymers to 400 °C induces the formation of archetypical chevron GNR **2**.

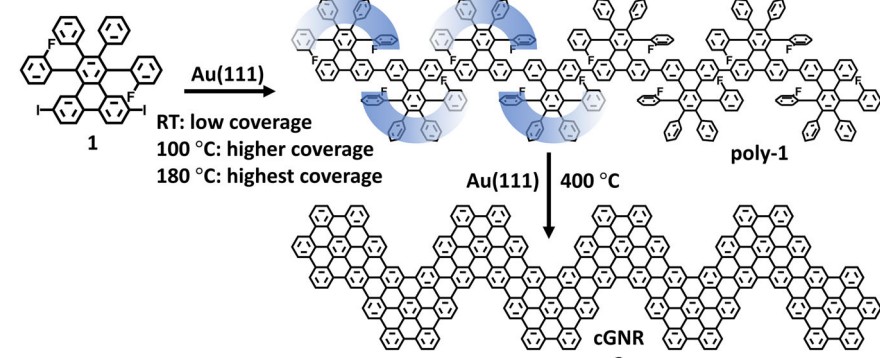

**Fig. 2 | Given equal deposition duration and crucible temperature, adsorption of precursor 1 onto the Au(111) surface and subsequent growth into polymers is dictated by substrate temperature during deposition.** All samples were created with identical deposition durations and crucible temperatures, with the only variation being in substrate temperature during deposition. **a, c, e** Large-scale and **b, d, f** small-scale scans of polymer assemblies from depositions performed at 22, 100, and 180 °C substrate temperatures, respectively. The green circle in **b** indicates a bright spot due to iodine that remains on the surface. Scan parameters: **a** -1.95 V, 30 pA; **b** 1.9 V, 10 pA; **c** -0.75 V, 30 pA; **d** -0.25 V, 10 pA; **e, f** -1.5 V, 15 pA.

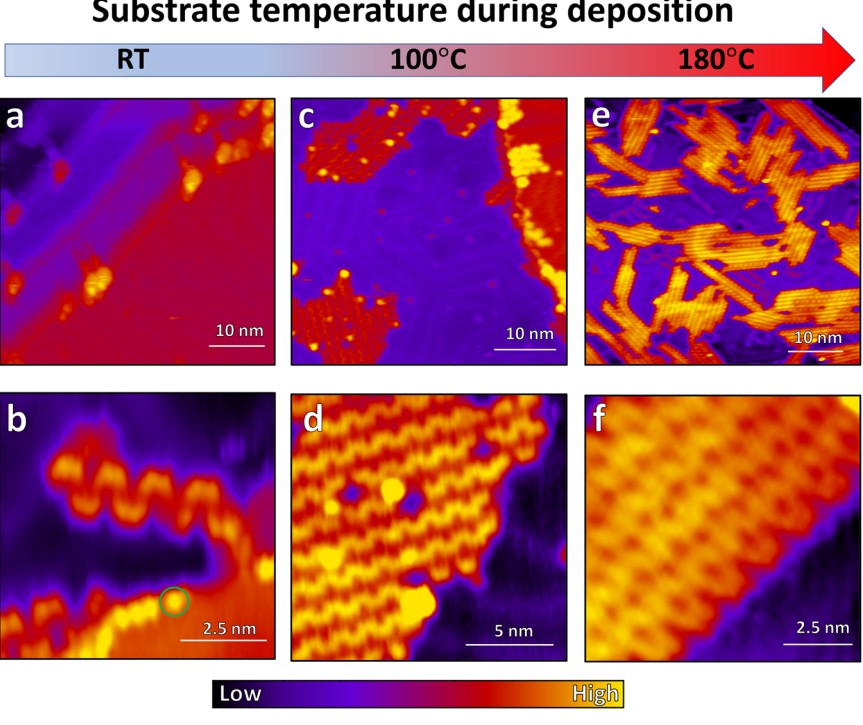

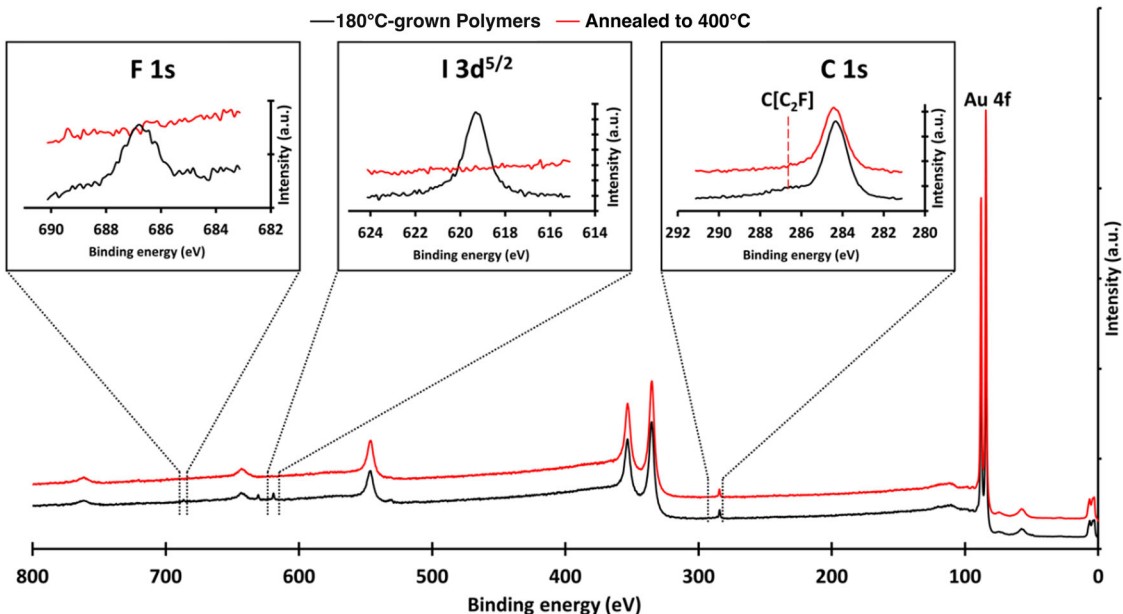

**Fig. 3 | X-ray photoelectron spectroscopy data from a 180 °C-grown polymer sample (black) and after annealing to 400 °C (red), with individual elemental spectra shown in inset.** The peaks corresponding to fluorine and the C[C2F] moiety are present after the initial polymer growth but disappear following cyclodehydrogenation. The entire 400 °C spectrum was offset vertically from the as-grown sample data for enhanced visual clarity.

the profiles (Supplementary Fig. 11g, h). Further investigations are needed to either identify the M-conformation or justify its absence. Additionally, the helical signatures are found more pronounced under lower bias condition and become diminished when a higher bias magnitude is applied (Supplementary Fig. 10). Evidence from previous studies showed that asymmetric bright spots in STM images appeared at lower bias conditions[11,16].

To better understand the effect of the growth temperatures, we now compare the growth of GNRs derived from the samples deposited at different temperatures. Figure 5 shows example STM images of the 180 °C (Fig. 5a) and 100 °C (Fig. 5c) samples before heating to 400 °C for cyclodehydrogenation plus cyclodehydrofluorination. Figure 5b, d show that the greatest average GNR length, greatest individual GNR length, and greatest total GNR coverage are obtained at the 180 °C polymerization temperature, with both average and greatest individual lengths reaching nearly double that observed at the 100 °C polymerization temperature. To shed light on this difference, we carried out DFT energy calculations for fluorine-bearing polymers adsorbed on Au(111) surface (optimized structures available in Supplementary Data 1). The fluorine-bearing polymer with C-F bonds pointing up ($F_{up}$) is found to have a negative adsorption energy of −2.96 eV, whereas the

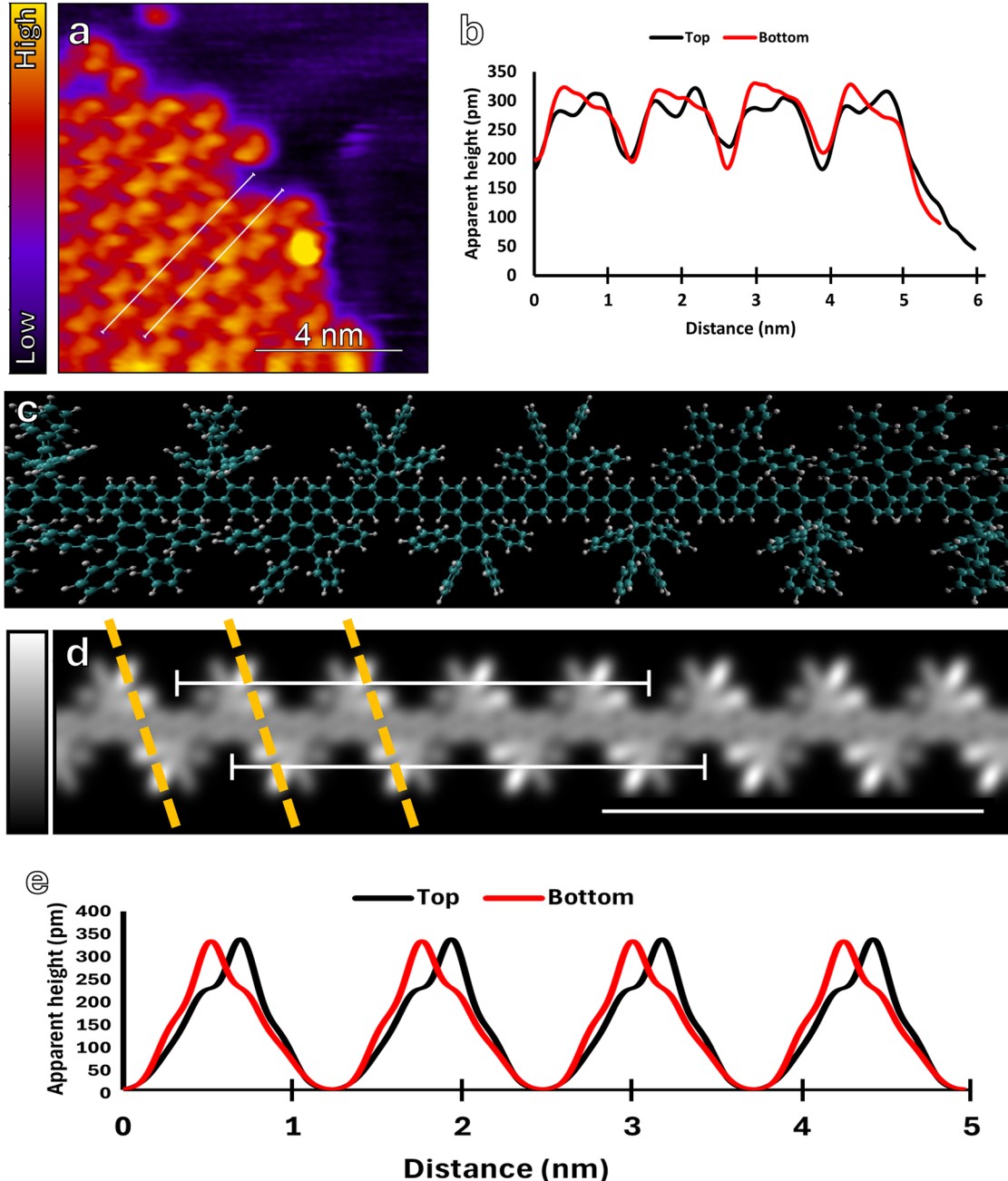

**Fig. 4 | Helicity in polymer intermediates. a** STM image of fluorinated chevron polymers acquired at −0.3 V, 55 pA. **b** Height profile under the two white lines in **a**, demonstrating the characteristic asymmetric appearance of the phenyl fringes on two sides along a representative polymer chain. The top trace exhibits a left peak lower in intensity than the right, while the bottom trace exhibits a left peak higher than the right. This is in good agreement with polymers with P-type helicity, shown in optimized geometry in **c**, where the polymer is given in a perspective view without Au substrate for clarity. **d** is a simulated p-type polymer along which traces **e** have been drawn as in **a**, revealing the same intrinsic peak shape observed in the experimental height profiles in **b**. Yellow lines have been added as a guide to the eye for the location within each subunit that exhibits the greatest apparent height. The scale bar in the bottom-right of **d** is 4 nm to coincide with **a**.

other configuration with C-F bonds pointing down ($F_{dn}$) has a positive adsorption energy of +0.16 eV, per unit cell of polymer. Due to the internal ortho-positions of F substitution, the preferred adsorption of $F_{up}$ over $F_{dn}$ is opposite to the previously reported trend for F atoms located on the polymer/GNR edges[14,33]. This large difference in adsorption strengths may contribute to the reduced fraction of surface coverage experimentally observed at lower temperature (Fig. 2). On the other hand, the adsorption energy of pristine polymers is also found to be negative at −0.10 eV, which is however weaker than the fluorine-bearing polymers in the $F_{up}$ configuration, due to the same interfacial contact between C-H bonds and Au substrate, and the smaller molecular

weight and accordingly weaker van der Waals interactions. Consequently, the stronger adsorption and larger molecular weight of the F-bearing precursors in the $F_{up}$ configuration dictate a higher temperature needed for the precursors to overcome the barrier and to diffuse on the Au(111) surface. Previous experimental studies have shown that the polymerization of chevron-type monomers is controlled by diffusion[42]. Obviously, the temperature is not too high to dissemble the 2D polymer islands stabilized by inter-digited π-π interactions. We conclude from these results that the higher temperature of the surface is more conducive to the diffusion of molecules on contact with the terraces, affording longer and more plentiful polymers and thus GNRs.

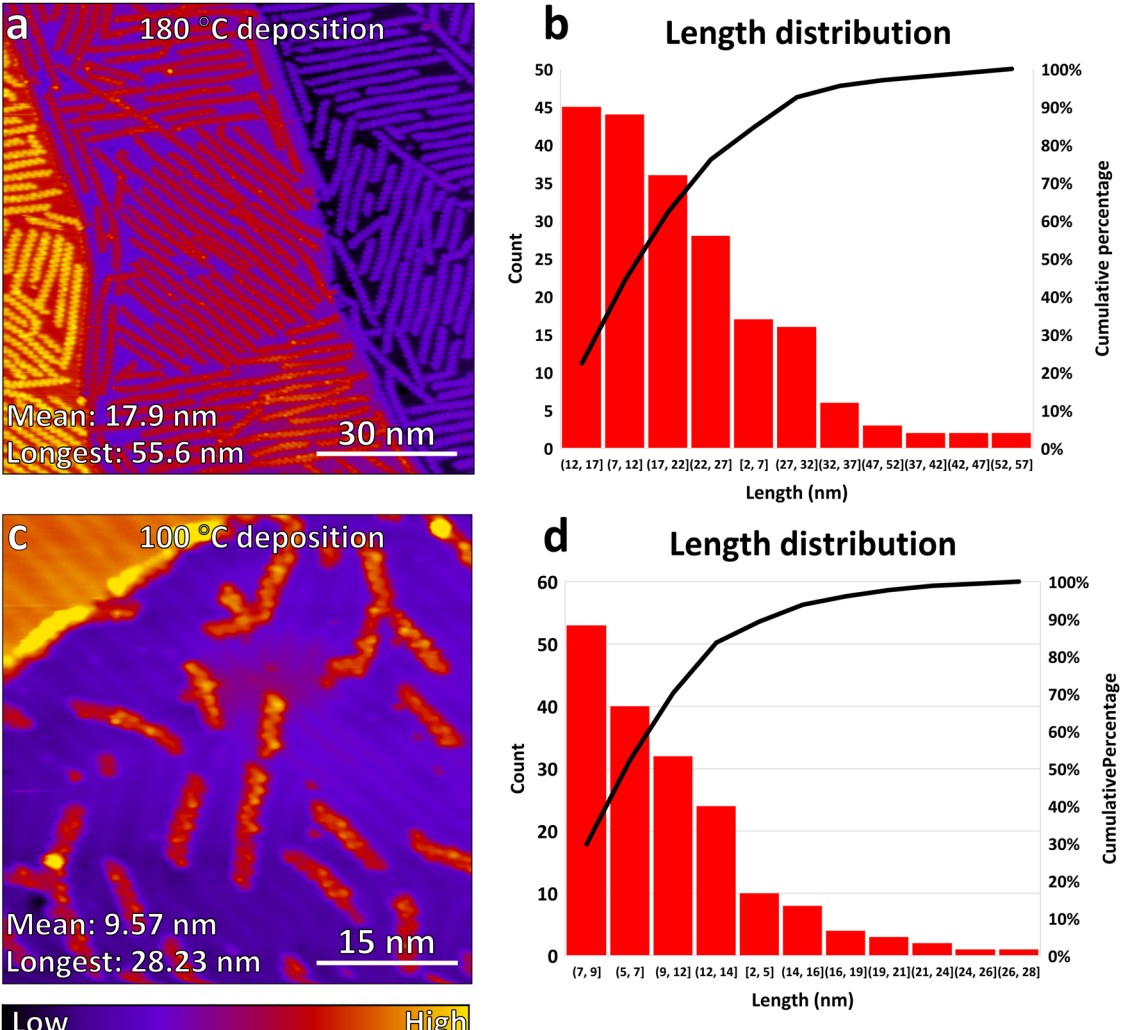

**Fig. 5 | Statistical data on GNR lengths derived from samples created with different surface temperatures during deposition. a** Large-scale scan of GNRs grown from precursors deposited at 180 °C substrate temperature. Scan parameters: −1.75 V, 25 pA. Scale bar 30 nm. **b** Pareto plot of GNR lengths from the sample shown in **a**, with the most populated bin ranging from 12 to 17 nm, mean length 17.9 nm, and longest individual GNR at 55.6 nm. **c** Representative scan of GNRs grown from precursors deposited at 100 °C substrate temperature. Scan parameters: −1.95 V, 20 pA. Scale bar 15 nm. **d** Pareto plot of GNR lengths from the sample shown in **c**, with the most populated bin ranging from 7 to 9 nm, mean length 9.57 nm, and longest individual GNR at 28.23 nm.

## Conclusions

The fluorine-bearing precursors $C_{42}H_{24}F_2I_2$ are shown capable of growing high-quality chevron GNRs on Au(111), but the adsorption and self-assembly of these precursors are heavily dictated by the substrate temperature during deposition. Whereas the pristine chevron precursor can be deposited in appreciable amounts at room temperature via sublimation[42] or even direct contact transfer[43], the fluorinated 2I-2F precursor is not observed to adsorb upon the Au(111) surface at lower temperatures. The fraction of sublimated molecules that adsorb on the Au(111) surface (and consequently the maximum possible polymer/GNR length) increases with increasing substrate temperature until approximately 180 °C, after which no appreciable marginal improvement is observed. We attribute the difference in adsorption behavior to the included fluorine atoms, as the pristine chevron precursor (with either bromine or iodine as a halogen group) is able to adsorb on room-temperature Au(111) without issue. Polymer intermediates are shown to be helically shaped, however the helical structure is not retained in the final chevron-type GNR products. This transient feature may still have implications for the synthesis of other helically shaped materials on surfaces.

## Materials/methods

### Graphene nanoribbon growth

The Au(111) single crystal was cleaned by repeated cycles of Ar+ sputtering and annealing at 740 K. Molecular precursor **1** was degassed at 170 °C overnight in a Knudsen cell. Detailed synthetic procedures of the precursor are described in the Supplementary Information. Deposition was performed with a crucible temperature of 200 °C. Polymerization appeared to occur sporadically at room temperature, but isolated precursors were never observed, nor were polymers on terraces. Reliable polymerization was achieved by annealing at 100 °C for 10 min (holding that temperature during precursor deposition), and full cyclization could be achieved by subsequently annealing at 400 °C for 15 min to yield chevron GNR **2**. Increasing the substrate temperature to around 180 °C during and after deposition significantly increased the proportion of adsorbed molecules as well as the average polymer (and thus GNR) length.

### Scanning tunneling microscopy

STM characterization was performed with a homemade variable-temperature system at 110 K under UHV conditions with a clean, commercially available PtIr tip. All STM images were acquired in

constant-current mode. The bias voltage was applied to the sample bias with respect to the tip.

## X-ray photoelectron spectroscopy

XPS measurements were made in ultrahigh vacuum chamber with an operating pressure of $2 \times 10^{-10}$ torr. X-rays were generated with a SPECS XR50M source using a Mg anode operated at 15 kV and 400 mA electron current. X-rays were then passed through a SPECS Focus 500 monochromator to narrow the photon energy distribution and remove satellites. Photoemitted electrons from the sample were analyzed in a SPECS Phoibos 150 hemispherical analyzer operated at 40 eV pass energy with lenses set for Medium Area. The sample normal was oriented toward the electron detector.

## Density functional theory calculations

Atomic structures were optimized with the massively parallel real-space multigrid (RMG) code [https://github.com/RMGDFT/rmgdft]. The RMG parallelizes over nodes, using all CPU cores and all GPUs in each node. The calculations used the PBE exchange-correlation functional[46] and van der Waals nonlocal correlation correction[47]. The real space grids were chosen so that the equivalent plane wave cutoff energies were 100 Ry for the wave functions and 400 Ry for the charge density. Adsorption energies were calculated by taking the difference between the total energy of the polymers adsorbed on the Au(111) surface and the sum of the total energies of the free polymers and Au(111) surface. The STM images were calculated with Tersoff-Hamann scheme[48] at constant current mode.

## Data availability

Supplementary Data 1 contains atomic coordinates of the optimized pristine and fluorinated polymer intermediates adsorbed on Au(111) surface. Correspondence and requests for materials should be addressed to J.H. or A.-P.L.

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

## Acknowledgements

This research was conducted at the Center for Nanophase Materials Sciences (CNMS), which is a DOE Office of Science User Facility. The electronic characterization was funded by ONR grants N00014-10-1-2302. The synthesis of the GNR precursors was supported by the ONR via N00014-19-1-2596. The supercomputer time was provided by DOE at the Oak Ridge Leadership Computing Facility and at the National Energy Research Scientific Computing Center (Contract No. DE-AC02-05CH11231 using NERSC award BES-ERCAP0027465).

## Author contributions

J.D.T. and M.S. contributed equally to this work. A.-P.L. conceived the project and designed the experiments; J.H. designed the theory tasks. J.D.T., C.T., A.P.B., and K.H. performed on-surface synthesis and characterizations; M.S. and A.S. conducted molecule synthesis; W.L., J.H., and J.B. performed the theoretical calculations. J.D.T., W.L., J.H. and A.-P.L. wrote the paper with contributions from all authors.

## Competing interests

The authors declare no competing interests.
