## [Peer Review File · Communications Chemistry]

Reviewers' comments:

Reviewer #1 (Remarks to the Author):

The authors claim in the title and abstract that they have developed the temperature-mediated adsorption and growth of helical polymers on gold from a novel internally fluorinated precursor. These fluorinated precursors are of interest to the community as they could facilitate the growth of graphene nanoribbons on non-metal substrates. The authors conclude by noting that, for their precursor, the maximum polymer length increases with deposition on increasingly hot substrate to a temperature up to around 180 degrees, while there is limited adsorption nor growth observed at room temperature. They also emphasize that the observed helical structure in the polymer is significant and may have implications for synthesizing other helical materials on surfaces.

The observation that precursors are not detected upon room temperature deposition but instead polymers are formed at higher substrate temperatures during deposition is a crucial aspect of this work, justifying its publication in Communications Chemistry, especially when a new functional precursor is considered.

That said, the assertion regarding helical polymers may not be fully supported by existing terminology. An internet search on "helical polymers" reveals vastly different structures such as staircase-like polymers, twisted polymers, or self-assembled structures. In this context, the helicity observed in the polymer may be better described as polymers with helical side-groups or similar, to avoid misleading the readers. Additionally, within this claim of partly helical polymers, the presence of several isomers is expected, yet not thoroughly addressed with different panels. Furthermore, the simulations presented do not illustrate the underlying gold and adsorption sites, making it unclear to what extent the simulated STM images or data are influenced by gold adsorption. Therefore, it is strongly recommended that the emphasis on "helical polymers" in the claim and conclusion is downplayed, while highlighting the significant finding of adsorption and polymerization above room temperature as the primary focus of the manuscript.

Minor details:

o Consider a schematic showing the major claims of the article as Figure 1/scheme 1.

Reviewer #2 (Remarks to the Author):

This manuscript by Jacob D. Teeter et al. presents a comprehensive STM experimental work aimed at kinetics optimization of on-surface, Ullmann-like, covalent coupling of a new, model fluorine-bearing precursor (C₄₂H₂₄F₂I₂) to obtain high-quality chevron-type GNRs on Au(111). The STM results are supported by X-ray photoelectron spectroscopy and DFT simulations.

The recent literature results [Fan et al. (2021). Science 372(6544): 852-856. and refs 37-38] demonstrated that selective HF elimination is a powerful strategy for nanographene materials formation on noble metals and metal oxides, respectively. In both cases, the literature on understanding the growth mechanism from fluorine-bearing precursors is limited. In this context, as the work is technically solid, I recommend it for publication after minor changes.

Before publication the authors may consider more detailed comments presented below.

1. The desorption of precursors with such high molecular mass is intriguing mainly since the synthesis of longer polymers is performed during the deposition of these precursors on the sample kept at even higher temperatures. As STM is a local technique, how are authors sure that the precursors do not agglomerate at RT in a more oversized island separated by distances comparable to or larger than STM scanning areas? Was XPS done on those samples showing no or minimal signals from precursors?

2. In the context of the above question, the DFT discussion from lines 185-190 mentioning more considerable adsorption energy from F-bearing molecular systems (with respect to pristine ones) is an argument against their preferential desorption at RT.

3. Line 52. The authors may expand their discussion by mentioning intermolecular coupling via HF elimination on Au(111) (ref. Fan et al. above).
4. STM height in experimental figures may be present in qualitative scale.
5. Fig. 3 top: the "RT" abbreviation is misleading as these polymers were not obtained at room temperature.

Point-by-point responses to Reviewer 1's comments

General comment: *The authors claim in the title and abstract that they have developed the temperature-mediated adsorption and growth of helical polymers on gold from a novel internally fluorinated precursor. These fluorinated precursors are of interest to the community as they could facilitate the growth of graphene nanoribbons on non-metal substrates. The authors conclude by noting that, for their precursor, the maximum polymer length increases with deposition on increasingly hot substrate to a temperature up to around 180 degrees, while there is limited adsorption nor growth observed at room temperature. They also emphasize that the observed helical structure in the polymer is significant and may have implications for synthesizing other helical materials on surfaces.*

The observation that precursors are not detected upon room temperature deposition but instead polymers are formed at higher substrate temperatures during deposition is a crucial aspect of this work, justifying its publication in Communications Chemistry, especially when a new functional precursor is considered.

Author reply: We thank Reviewer 1 for carefully reviewing our manuscript, providing positive evaluations, and recommending it for publication in Communications Chemistry. Please find our further responses to Reviewer 1's additional comments below.

Comment 1: *That said, the assertion regarding helical polymers may not be fully supported by existing terminology. An internet search on "helical polymers" reveals vastly different structures such as staircase-like polymers, twisted polymers, or self-assembled structures. In this context, the helicity observed in the polymer may be better described as polymers with helical side-groups or similar, to avoid misleading the readers.*

Author reply: We agree with Reviewer 1 on this comment and suggestion. We report the observation of helical conformation for the polymer intermediates on Au(111) surface for the first time. The observation is based on joint experimental and theoretical characterizations. However, the helical conformation of the polymer intermediates differs from real helical polymers, see, e.g., the helical polymer that some of us (Hong and Li) reported in a separate study: Zhang, H.-H., et al. Helical Poly(5-alkyl-2,3-thiophene)s: Controlled Synthesis and Structure Characterization, *Macromolecules* **2016**, *49*, 4691-4698. For clarity, we have followed Reviewer 1's suggestion to change "helical polymers" to "helically shaped polymers" in the title and throughout the manuscript, and specified that "polymers in fact exhibit distinct helical signatures on the phenyl side groups", and we have also changed the word "configuration" to "conformation" everywhere.

Comment 2: *Additionally, within this claim of partly helical polymers, the presence of several isomers is expected, yet not thoroughly addressed with different panels.*

Author reply: We agree that a more complete discussion of the different conformations of helically shaped polymers would be helpful to readers. Therefore, between pages 8 to 9, we have expanded the discussions on helicity by referring to different panels of Fig. S11 as follows:

“The precursor **1** can polymerize into chains with four possible helicities, including P- and M-, meso- and nonhelical conformations (Fig. S11). By comparing the experimental line profiles with those generated from our DFT calculations, we find that the observed polymer intermediates belong to a P-conformation (Fig. S11a,b) instead of an M-conformation (Fig. S11c,d). In contrast, meso-chains would show asymmetric high-low feature for both traces (Fig. S11e,f), while nonhelical chains would show two peaks of equal heights in the profiles (Fig. S11g,h).”

Comment 3: *Furthermore, the simulations presented do not illustrate the underlying gold and absorption sites, making it unclear to what extent the simulated STM images or data are influenced by gold absorption. Therefore, it is strongly recommended that the emphasis on "helical polymers" in the claim and conclusion is downplayed, while highlighting the significant finding of adsorption and polymerization above room temperature as the primary focus of the manuscript.*

Author reply: We agree that Fig. S10 presents height profiles for polymers located on Au(111) surface, whereas Fig. S11 shows height profiles of isolated polymer chains with different helicities. To include van der Waals interactions between the polymers and the Au(111) surface, we have presented in Fig. S12 the simulated STM images of helically shaped fluorine-bearing polymers placed on Au(111) surface. It can be seen that the polymers retain their helical shape as indicated by the bright and dark features.

Nevertheless, we have downplayed the significance of helicities by making the following changes in the conclusion section:

“Polymer intermediates are shown to be helically shaped, however the helical structure is not retained in the final chevron-type GNR products. This transient feature may still have implications for the synthesis of other helically shaped materials on surfaces.”

Minor details: *o Consider a schematic showing the major claims of the article as Figure 1/scheme 1.*

Author reply: We have changed 100°C in Figure 1 to three rows:

RT: low coverage

100 °C: higher coverage

180 °C: highest coverage

and Figure 1 caption is updated to:

“Fluorine-bearing iodinated precursor **1** forms helically shaped polymers (**poly-1**) on Au(111) with increasing coverages from room temperature (RT) to 100°C and 180°C.”

=====

Point-by-point responses to Reviewer 2's comments

General comment: *This manuscript by Jacob D. Teeter et al. presents a comprehensive STM experimental work aimed at kinetics optimization of on-surface, Ullmann-like, covalent coupling of a new, model fluorine-bearing precursor (C42H24F2I2)) to obtain high-quality chevron-type GNRs on Au(111). The STM results are supported by X-ray photoelectron spectroscopy and DFT simulations.*

The recent literature results [Fan et al. (2021). *Science* 372(6544): 852-856. and refs 37-38] demonstrated that selective HF elimination is a powerful strategy for nanographene materials formation on noble metals and metal oxides, respectively. In both cases, the literature on understanding the growth mechanism from fluorine-bearing precursors is limited. In this context, as the work is technically solid, I recommend it for publication after minor changes.

Before publication the authors may consider more detailed comments presented below.

Author reply: We thank Reviewer 2 for recommending our manuscript for publication and pointing us toward a reference that is closely relevant to our work involving HF-elimination. We have added this paper as a new reference 34 on page 3 and revised relevant sentence (see our response to Comment 3 below for details).

Comment 1: The desorption of precursors with such high molecular mass is intriguing mainly since the synthesis of longer polymers is performed during the deposition of these precursors on the sample kept at even higher temperatures. As STM is a local technique, how are authors sure that the precursors do not agglomerate at RT in a more oversized island separated by distances comparable to or larger than STM scanning areas? Was XPS done on those samples showing no or minimal signals from precursors?

Author reply: The desorption of precursors is indeed intriguing and puzzling - On the one hand, the precursors do not seem to adsorb at RT, and on the other hand, longer polymers are synthesized at higher temperature. Unfortunately, we have not measured XPS for the RT grown polymer sample, as our STM measurements, despite not exhaustive, don't show any polymers or islands/agglomerations of precursors. We believe that the apparent desorption of precursors most likely results from the lower sticking coefficient of the fluorinated precursors at RT. To verify this point, we have carried out additional calculations to compare adsorption energies for the fluorinated polymers with C-F bonds pointing up and down (F_{up} and F_{dn}) and the pristine polymers. The adsorption energies are found to be -2.96 eV, +0.16 eV, and -0.10 eV per unit cell of polymer, respectively for F_{up} , F_{dn} , and pristine polymers. It is noted that F_{dn} has a positive adsorption energy, meaning the molecules with C-F pointing down do not adsorb on the surface. In comparison, pristine precursors have negative adsorption energy, which is consistent with the observation that the pristine precursors can stay on the surface forming polymers at RT (Fig. S8). Now why is higher temperature better than lower temperature for fluorinated polymers? This is probably because higher temperature is conducive to diffusion of F_{up} (remember F_{dn} does not adsorb) that is essential for polymerization. Indeed, earlier studies have shown that the polymerization of chevron-type monomers is diffusion controlled [see Bronner, C. *et al.* Iodine versus Bromine Functionalization for Bottom-Up Graphene Nanoribbon Growth: Role of Diffusion. *Journal of Physical Chemistry C* **121**, 18490-18495 (2017)]. Once polymers form, they can stay together by interdigitated pi-pi interactions. Therefore, the polymerization that begins at higher temperatures facilitates the retention of molecules through interchain pi-pi interactions.

We have clarified this point in the manuscript on Pages 11-12.

Comment 2: In the context of the above question, the DFT discussion from lines 185-190 mentioning more considerable adsorption energy from F-bearing molecular systems (with respect to pristine ones) is an argument against their preferential desorption at RT.

Author reply: This is a good point. As explained above, our additional calculations show adsorption energies change in the order of $F_{up} > H > F_{dn}$. The positive adsorption energy F_{dn} means the molecules with C-F pointing down do not adsorb on the surface, which can explain the reduced surface coverage at RT for the fluorinated precursor, as compared to the pristine chevron molecular precursor (Fig. S8).

The stronger adsorption and larger molecular weight of the F-bearing precursors in the F_{up} configuration require a higher temperature for the precursors to overcome barrier to diffuse on the Au(111) surface since the polymerization of chevron-type monomers is controlled by diffusion [Ref. 42]. Once the precursors are deiodinated and polymerized, they can form pi-pi interaction stabilized island at higher temperature. The elevated temperature is just more conducive to the diffusion of molecules on contact with the terraces, affording longer and more plentiful polymers and thus GNRs.

Comment 3: *Line 52. The authors may expand their discussion by mentioning intermolecular coupling via HF elimination on Au(111) (ref. Fan et al. above).*

Author reply: We have followed the suggestion to add a new reference 34 and revised the relevant sentence to:

"... but also to facilitate inter- and intra-molecular aryl-aryl coupling through HF elimination on both metal and metal oxide surfaces^{34,35}."

Comment 4: *STM height in experimental figures may be present in qualitative scale.*

Author reply: STM heights in experimental images (Fig. 2) are indicated in a qualitative (but linear) low-high scale, and more quantitative values can be seen in the line scans of Fig. 4.

Comment 5: *Fig. 3 top: the "RT" abbreviation is misleading as these polymers were not obtained at room temperature.*

Author reply: We agree with the reviewer and have corrected "RT Polymers" in Fig. 3 to "180°C-grown Polymers".

REVIEWERS' COMMENTS:

Reviewer #1 (Remarks to the Author):

Prior to publication:

Consider rendering the DFT structures in Figure S12 in a manner whereby information can be extracted (e.g. visualization from the side, from the top, showing adsorption height and other typical distances) and uploading the raw data.

Reconsider the title and message throughout: The statement "helically shaped polymers" remains somewhat misleading, whereas the phenomenon could be better described as "chiral adsorption" or similar. Likewise "temperature-mediated growth" appears to describe the growth method for most on-surface fabricated materials and might not convey novelty. Instead, for example "Deposition-temperature growth of chiral polymers and graphene nanoribbons from an internally fluorinated precursor" might accurately convey the results and impact of the work.

Reviewer #2 (Remarks to the Author):

The authors satisfactorily responded to my suggestions. I recommend the work for publication.

Point-by-point responses to Reviewer #1's comments

Comment 1: *Consider rendering the DFT structures in Figure S12 in a manner whereby information can be extracted (e.g. visualization from the side, from the top, showing adsorption height and other typical distances) and uploading the raw data.*

Author reply: We thank the Reviewer for the valuable suggestion.

Fig. S12 is shown from side in panels a,d,g, and from top in panels b,c,e,f,h,i. Since both the polymer intermediates and the Au(111) surface become non-planar after relaxation, the adsorption height is not a well-defined quantity. Following the Reviewer's suggestion, we have uploaded in a new document "Supplementary Data 1" the raw data in terms of lattice vectors and atomic coordinates for the three polymer intermediates adsorbed on Au(111) surface.

Comment 2: *Reconsider the title and message throughout: The statement "helically shaped polymers" remains somewhat misleading, whereas the phenomenon could be better described as "chiral adsorption" or similar. Likewise "temperature-mediated growth" appears to describe the growth method for most on-surface fabricated materials and might not convey novelty. Instead, for example "Deposition-temperature growth of chiral polymers and graphene nanoribbons from an internally fluorinated precursor" might accurately convey the results and impact of the work.*

Author reply: We thank the Reviewer for the valuable suggestions. We have added "Deposition" at the beginning of the title to convey novelty and removed "on Au(111)" at the end, as it is already mentioned in the abstract. However, we believe it is better to retain the words "helically shaped" instead of changing them to "chiral". The term of chirality is too broad and refers to non-superimposable mirror images, whereas in this work we deal with helically shaped polymers exhibiting chirality along the polymer axis. This type of axial chirality is essentially helicity. As the Reviewer noted in the previous review, the polymer intermediates we observed differ from other helical polymers. Nonetheless, we believe they should still be described as helically shaped, due to the sinusoidal ups and downs of the phenyl side groups along the polymer backbone axis. Therefore, we propose the title "Deposition temperature-mediated growth of helically shaped polymers and chevron-type graphene nanoribbons from a fluorinated precursor". Please note that we also changed "an internally fluorinated" to "a fluorinated" to meet the word count requirement. We hope the Reviewer agrees with our rationale and choice of title.

=====

Point-by-point responses to Reviewer #2's comments

General comment: *The authors satisfactorily responded to my suggestions. I recommend the work for publication.*

Author reply: We thank the Reviewer for recommending our manuscript for publication.